# Iodine Nutritional Status and Related Factors among Chinese School-Age Children in Three Different Areas: A Cross-Sectional Study

**DOI:** 10.3390/nu13051404

**Published:** 2021-04-22

**Authors:** Xiaoyun Shan, Changqing Liu, Xiaoyan Luo, Yan Zou, Lichun Huang, Weiwen Zhou, Qiulan Qin, Deqian Mao, Min Li, Lichen Yang

**Affiliations:** 1Chinese Center for Disease Control and Prevention, National Institute for Nutrition and Health, Key Laboratory of Trace Element Nutrition, National Health Commission of the People’s Republic of China, 29 Nanwei Road, Beijing 100050, China; shanxiaoyun0924@163.com (X.S.); maodq@ninh.chinacdc.cn (D.M.); limin@ninh.chinacdc.cn (M.L.); 2School of Public Health, University of South China, 28 Changsheng West Road, Hengyang 421001, China; 3Hebei Provincial Center for Disease Control and Prevention, Shijiazhuang 050011, China; lcq93@126.com (C.L.); hebeishipinsuo@163.com (X.L.); 4Zhejiang Provincial Center for Disease Prevention and Control, 3399 Binsheng Road, Hangzhou 310051, China; yzou@cdc.zj.cn (Y.Z.); lchuang@cdc.zj.cn (L.H.); 5Guangxi Center for Disease Prevention and Control, 18 Jinzhou Road, Nanning 530028, China; wwzhou2000@126.com (W.Z.); qinqiulangxcdc@163.com (Q.Q.)

**Keywords:** UIC, TSH, vitamin A, vitamin D, school-age children

## Abstract

We evaluated the iodine nutritional status and related factors among school-age children based on the 2016 National Nutrition and Health Surveillance of Children and Lactating Women; 3808 children from Hebei, Guangxi, and Zhejiang province were included in the study. Urinary iodine concentration (UIC), thyroid-stimulating hormone (TSH), body mass index (BMI), vitamin A (VA), and vitamin D (VD) were measured. The abnormal rate of UIC and TSH were assessed. Relationships between UIC/TSH and the possible factors were analyzed. The overall median UIC was 185.14 µg/L, and the median UIC of children aged 8–10 was 164.60 µg/L. Prevalence of iodine deficiency and excess was 13.84% and 14.36%, respectively, and 12.87% of children showed TSH excess. UIC, as well as the abnormal rates of iodine deficiency (ID) and TSH, were significantly different among the three provinces. The median UICs and excess rates increased with age, reaching 211.45 µg/L and 21.35% at age of 14~, while TSH showed the opposite trend. Overweight children tended to have lower UIC and higher TSH. Higher UIC and TSH were found in VA sufficient group (*p* < 0.01). Further, the VD deficient group had a higher TSH compared to the sufficient group (*p* < 0.01). Moreover, UI and TSH distribution was obviously different among different vitamin A/D status (*p* < 0.05). Although the median UIC of school-age children was optimal, there were pockets of inadequate and excessive UI in the three provinces. Compared to the national IDD monitoring results in 2014, the iodine nutritional status of children was greatly improved. Considerations of region, age, BMI, VA, or VD are needed in the future iodine evaluation and surveillance.

## 1. Introduction

Iodine, a trace element essential for thyroid hormone synthesis, is mainly obtained from diet and excreted in urine. Inadequate iodine intake results in a spectrum of iodine deficiency disorders (IDD), including thyroid disorders, developmental, and functional morbidities from the fetal stage to adulthood [1]. ID in childhood causes impaired cognitive development and growth retardation and is the most common cause of preventable mental impairment worldwide [2]. However, excessive iodine intake may also contribute to nodular goiter, hyperthyroidism and Hashimoto’s thyroiditis development [3]. Moreover, ‘*The Monitoring of Iodine Deficiency Disorders of China in 2014*’ reported that a proportion of nearly 20% children showed iodine excess [4]. Thus, continuous monitoring of the iodine status of the susceptible population, such as the pregnant women, infants and children, is quite important.

World Health Organization/International Council for Control of Iodine Deficiency Disorders/United Nations International Children’s Emergency Fund (WHO/ICCIDD/UNICEF) recommend UIC as one of the indicators for evaluating population iodine nutrition [5]. Moreover, the UIC of children aged 8–10 is usually used to reflect the iodine nutritional status of the general population [6]. According to the Global Scorecard of Iodine Nutrition in 2019, about 0.25 million schoolchildren are at risk of iodine deficiency worldwide [7]. In China, iodine deficiency or excess (median UIC < 100 µg/L or ≥ 300 µg/L respectively) still exists in some provinces or regions. There were 19 provinces with iodine deficiency rate > 15%, and 17 provinces and corps with iodine excess rate > 15%. Moreover, eastern China had the lowest proportion of adequate iodine nutrition [4].

The WHO has also suggested that populations are classified as iodine sufficient if the proportion of newborn TSH > 5 mIU/L in the population is < 3% [5]. Moreover, the TSH marker has a better agreement with goiter prevalence than median UIC when classifying the iodine status of populations [8]. VA has been shown to inhibit the secretion and synthesis of TSH [9], and the nutritional status of VA has been significantly improved in children and adolescents aged 6 to 17 years in China (prevalence of VA deficiency = 0.96%) [10]. However, VD deficiency is still a serious health problem in children and adolescents. The general prevalence of VD deficiency was 53.2% in 2010–2012 [11]. A population-based study showed that high VD status in younger individuals was associated with low circulating TSH [12]. Interactions of vitamin A/D and iodine deficiencies on thyroid function were also reported [13,14], and evidence suggests that thyroid dysfunction can easily lead to obesity, and vice versa [15]. Moreover, a recent study for the first time demonstrates that BMI may be a confounding factor in monitoring iodine nutritional status in schoolchildren [16].

Therefore, this study aims to access the current iodine nutritional status of school-age children in three representative provinces in China, aided by the 2016 National Nutrition and Health Surveillance of Children and Lactating Women. We expanded the age range of children and additionally measured the other variables, such as TSH, VA, and VD, to provide useful information for establishing better evaluation criteria in China and making monitoring policies in the future.

## 2. Materials and Methods

### 2.1. Study Design and Sample Collection

This cross-sectional study was conducted based on the 2016 National Nutrition and Health Surveillance of Children and Lactating Women. Hebei, Guangxi, and Zhejiang, located in the north, south, and coast of eastern China were selected. In each province, five districts or counties were randomly selected from five different geological locations (areas with non-high water iodine). One primary or middle school was randomly chosen from each town. A total of 28 healthy children (14 boys and 14 girls) from one class in each grade were recruited by cluster sampling. In each district or county, 20 students were randomly selected from each age group (between 6 and 17 years). Those who have history of thyroid disease or confirmed using thyroid-related medication were not enrolled in the study. Written informed consent was obtained from the principal of the school and verbal informed consent from parents or legal guardians of participating children were normatively recorded. This study was approved by the Ethical Review Committee of Center for Disease Control and Prevention (CDC) and all procedures performed in studies involving human participants were in accordance with the ethical standards of the committee.

### 2.2. Anthropometric Measurements

Anthropometric measurements, including standing height (cm) and weight (kg), were taken by local health professionals according to a standard protocol. Height and weight were respectively measured to the nearest 0.1 cm and 0.1 kg with the subjects standing without shoes and wearing light clothing only. BMI was calculated as weight in kilograms divided by the square of height in meters (kg/m^2^). All participants were categorized into three groups of underweight, normal weight, overweight/obesity according to the BMI growth reference values for Chinese children and adolescents aged 6–18 years suggested by screening standard for malnutrition of school-age children (WS/T 456-2014) and adolescents and Screening for overweight and obesity among school-age children and adolescents (WS/T 586-2017).

### 2.3. Urinary Iodine, Creatinine, TSH, Retinol, and 25(OH)D Concentrations Determination

For each child, approximately 8–10 mL of a random spot midstream urine sample was collected in the morning from 08:30 to 12:00. Laboratories at the provincial level were approved after passing the examination of the National Iodine Reference Laboratory, China CDC. UIC was measured using arsenic and cerium catalysis spectrophotometry (WS/WS 107-2006). Urinary creatinine was measured by a national standard method (determination of urine creatinine with spectrophotometric Method, WS/T97-1996). Iodine nutritional levels <100 µg/L, 100–199 µg/L, 200–299 µg/L, and ≥ 300 µg/L were defined as insufficient, adequate, above adequate, and excessive, respectively.

Blood samples (6 mL) were collected from the cubital vein of children. Serum samples were prepared by centrifugation at 3000 rpm for 10 min after allowing the samples to stand for 30 min and were subsequently frozen at −80 °C until analysis. TSH levels were determined using an automated chemiluminescence immunoassay analyzer (Roche, German). The normal reference range (95% confidence interval [CI]) used for TSH according to the Roche Kit was as follows: 0.27–4.20 mIU/L. The serum retinol level was determined using high-performance liquid chromatography (WS/T 553-2017). When the measured serum (plasma) retinol content is less than the corresponding RV, it is determined as vitamin A marginal deficiency (<0.3 µg/mL) or deficiency (<0.2 µg/mL). The vitamin D level was determined using liquid chromatography-mass spectrometry (WS/T 677-2020), and a serum 25-hydroxyvitamin D (25(OH)D) concentration of 30 nmol/L (12 ng/L), 30–50 nmol/L (12–20 ng/L) or 50 nmol/L (20 ng/mL) is considered deficient, insufficient, or sufficient, respectively.

### 2.4. Statistical Analysis

Data were input in Microsoft Office Excel 2007 and data analysis was performed using IBM SPSS Statistics 23. Kolmogorov–Smirnov (KS) test was used for normality test. UIC and TSH were expressed as median and P_25_–P_75_. The Wilcoxon test and Kruskal–Wallis test were used to compare UIC and TSH among subgroups. A generalized linear model of the association between TSH and possible factors (UIC, VA, and VD) was built adjusting for region, age (continuous), sex, and BMI (continuous). Comparison of rates among subgroups was done using chi-squared test. *p* < 0.05 was considered statistically significant.

## 3. Results

### 3.1. Description of the Population

A total of 3808 schoolchildren, 1929 boys (50.70%) and 1879 (49.30%) girls, were included in the study. Characteristics of the study subjects are presented in Table 1. The median age and BMI of the participants were 11.32 (P_25_–P_75_: 9.00–13.97) years and 17.80 (P_25_–P_75_: 15.83–20.41) kg/m^2^, respectively. The overall median UIC was 185.14 (P_25_–P_75_: 129.60–252.88) μg/L, indicating that the iodine nutritional status of these children was adequate. Moreover, the P_25_–P_75_ values suggested that numbers of children showing VA, 25 (OH) D, TSH deficiency. Moreover, significant differences were observed with regard to BMI, UIC, UI/Cr, TSH, VA, 25 (OH) D among subjects from the three provinces (*p* < 0.01). Though no notable difference was found in UIC between Hebei and Zhejiang (*p* = 1.00), Hebei had a higher value than Zhejiang as well as Guangxi after adjusting by urinary creatinine (*p* < 0.01).

### 3.2. UI Concentrations and Distribution

Just as shown in Table 2, UIC is mainly concentrated in 100–299 μg/L. However, there were still a large number of children having UIC measurements below 100 μg/L (13.84%) and above 299 μg/L (14.36%). Guangxi had a significantly higher median UIC and abnormal rate than the other two provinces (210.10 μg/L and 33.62% respectively, *p* < 0.01). UIC was not statistically different between boys and girls (*p* = 0.17). Significant difference was observed among age groups (*p* < 0.01), and UIC increased gradually with age. More interestingly, for those children with normal BMI, the highest median UIC was discovered, and statistical difference was found between normal group and overweight group (*p* < 0.01). Meanwhile, the proportion of deficient UIC was also higher in the overweight group (*p* < 0.01). In addition, a relatively higher UIC was found in the VA sufficiency group (*p* < 0.01). The distribution of UIC among the VA and VD status subgroups were significantly different. For children aged 8–10, the median UIC was 164.60 (P_25_–P_75_:115.50–219.83) μg/L, with 17.51% of deficiency rate and 8.12% of excess rate.

### 3.3. Serum TSH Levels and Distribution

In these three provinces, median TSH was 2.36 (P_25_–P_75_: 1.64–3.28) mIU/L, and 12.79% of children showed TSH excess. The median TSH concentration, as well as the prevalence of elevated TSH, was significantly higher in children from Hebei and Guangxi (*p* < 0.01). Meanwhile, no significant difference was found between boys and girls (*p* = 0.21). We additionally observed a trend for decreased serum TSH concentrations with age in children over 8 years. Moreover, the proportion of TSH above RV in the 14~ group was the lowest (8.14%). In addition, overweight children were prone to have higher TSH concentration, as well as excess TSH (Table 3).

### 3.4. Characteristics of Children with Underweight, Normal Weight and Overweight/Obesity

Of the participants, the prevalence was 22.27% (848/3808) for general overweight/obesity (13.18% for boys and 9.09% for girls) based on BMI (Table 4). Overweight/obese boys had a higher serum TSH and VA concentration than normal weight boys (median TSH: 2.62 vs. 2.37 mU/L, *p* < 0.01; median VA: 0.39 vs. 0.36 μg/mL, *p* < 0.01), but serum UIC and 25 (OH) D levels showed no significant difference between the two groups (*P*_UIC_ = 0.11 and *P*_25 (OH) D_ = 0.06). Moreover, overweight/obese girls showed a lower UIC than normal weight girls (median UIC: 171.65 vs. 192.87 μg/L, *p* < 0.01). Moreover, overweight/obese boys had a higher serum TSH, VA, and 25 (OH) D concentration than overweight/obese girls (median TSH: 2.62 vs. 2.37 mU/L, *p* = 0.03; median VA: 0.39 vs. 0.37 μg/mL, *p* = 0.04; median 25 (OH) D: 22.27 vs. 19.08 ng/mL, *p* < 0.01).

### 3.5. Correlation between Urinary Iodine, VA, VD Status, and TSH Levels

Table 5 shows the association between TSH and UIC, vitamin A/D status. There was no association between TSH and UIC status even after adjusting for confounders. However, VA deficiency had a significant negative association with TSH (β = −0.597 mIU/L; 95% CI −0.868 to −0.326, *p* < 0.01). In contrast, VD deficiency could lead to remarkable TSH increasing (β = 0.609 mIU/L; 95% CI 0.437 to 0.781, *p* < 0.01).

### 3.6. Factors Related to TSH Distribution

We further analyzed the possible factors related to TSH distribution. The TSH abnormal rate in different UI, VA, and VD levels was shown in Figure 1. Only a small percentage of children had TSH levels below the RV (<0.27 mIU/L), and TSH distribution was not associated with UI status. The VA sufficient group showed a quite higher percentage (13.80%) above RV (>4.27 mIU/L). Moreover, children with VD deficiency had a higher percentage of excess TSH, compared to the sufficient group (OR = 1.931, 95% CI: 1.484–2.512, *p* < 0.01).

## 4. Discussion

In this study, the median UIC was 185.14 μg/L, indicating optimal iodine nutrition. Only 13.84% showed UICs < 100 μg/L and 14.36% ≥ 300 μg/L in the surveyed population, which were lower than that (15.8% and 20.6%, respectively) reported in 2014 [4]. Moreover, the median UIC and excess rate increased with age. This age-related characteristic has not been reported in previous studies, suggesting that we need to pay more attention to iodine excess in older children. The total median UIC of the children aged 8–10 years was 164.60 μg/L, consistent with the ‘High-Risk Endemic Survey’ result (162.60 μg/L) in 2014 [17]. For these children (with 8.12% of iodine excess), the proportion of ID accounted for 17.51%. This deficiency proportion has been historically higher, reaching 27.10% of the population in the 12 high-risk IDD provinces according to surveillance conducted from 2012 to 2014 [17]. Thus, our results demonstrated a continuous improvement of iodine nutritional status of Chinese children.

As there are many factors influencing the single urinary iodine, estimated 24 h UIE and UI/Cr are often used to evaluate the iodine intake in children and pregnant women [18,19]. In our study, both the UIC and UI/Cr values were obviously different among the three provinces, and after the adjustment with urinary creatinine, the differences of iodine status were further highlighted. In terms of considerable racial and age differences in creatinine excretion, we need further study to establish reference intervals of UI/Cr to evaluate iodine status for Chinese children.

In our population, 12.87% of the children had TSH levels > RV, and thus were at risk for developing clinical hypothyroidism. The median TSH concentration was significantly higher in children aged 8–10 and declined with age over 8 years in this study, while the median UICs increased with age. These results were inconsistent with the study by Johner et al. [20], which showed an association between higher iodine intakes and a shift in TSH towards higher levels in children. Though the TSH marker has a better agreement with goiter prevalence than median UIC when classifying the iodine status of populations [8] and studies have demonstrated a U-shaped relationship between UI and goiter prevalence [17], no notable correlation was found between UIC and TSH in our study. It may be that TSH is not a sensitive indicator for evaluating iodine nutrition in children.

According to the geographical and economic distribution, the overall level of iodine nutrition in the eastern region was lower than that in the central and western regions. Hebei, Zhejiang, and Guangxi in the eastern region used to be high-risk areas. In this study, the median UIC, as well as abnormal rate of ID, was the highest in Guangxi, and the median UI/Cr in Hebei was lowest. In addition, the median TSH and proportions of TSH below or above RV in Hebei and Guangxi were higher than that in Zhejiang. These differences may be related to economic development, eating habits, iodized salt consumption and water iodine content.

Ekinci et al. has proved that higher TSH levels is associated with overweight and obesity in children [21]. In the present study, 22.27% (848/3808) of children were classified as overweighted and obese (13.18% for boys and 9.09% for girls). Moreover, more children in this group tended to have ID and TSH excess compared to the group with normal body weight, and overweight/obese boys had the highest serum TSH and VA concentration, while overweight/obese girls had the lowest UIC. These results further indicated the effects of overweight and obesity on iodine nutritional status. A study from Isfahan reveals that VA deficiency and low VA status are not among the contributing factors of goiter persistence in the iodine replenished area [22]. However, VA supplements are effective in suppressing the pituitary TSHbeta gene in mild ID areas and can decrease excess TSH stimulation to the thyroid and thereby reduce the risk of goiter and its sequelae [23]. On the other hand, VD plays important roles in the pathogenesis of thyroid autoimmunity, and seasonal variations of VD could inversely influence levels of TSH [24]. In our study, the median UIC and TSH were the highest in children with sufficient VA. At the same time, these children seemed to be more prone to excessive UI and TSH. However, children with VD deficiency tended to have higher TSH excess. Since UIC, TSH and vitamin A/D are related to region, age, BMI, and other factors, we controlled the potential confounders when analyzing the relationship between TSH and UIC, vitamin A/D. We also found that VA sufficiency and VD deficiency were related to excess TSH, but UIC was not associated with vitamin A or D after controlling for region, age, sex, and BMI in these children, which needed to be further studied.

In summary, we used indicators including UIC, UI/Cr, and TSH to assess the iodine status and functions among children aged 6–17 from three different provinces, and all the results showed nice consistency. We also analyzed the effects of other factors such as age, BMI, VA, and VD on iodine nutritional status, which provided scientific data for comprehensive iodine assessment and surveillance in the future.

However, this study had some limitations: (1) free thyroxine (fT4), TPO-Ab (thyroid peroxidase antibodies), and TG-Ab (thyroglobulin antibodies) were not measured due to the limited blood samples available; (2) the data of dietary iodine including salt iodine were not detailed enough, which made it difficult to analyze the consistency of dietary iodine and urinary iodine.

## 5. Conclusions

In this study of school-age children from the three representative areas, we found decreased proportion of iodine deficiency and TSH excess compared to the national IDD monitoring results in 2014. Moreover, UIC and TSH were notably associated with region, age, BMI, VA, or VD. These findings provide useful information for establishing better iodine nutritional evaluation and surveillance criteria in China in the future.

## Figures and Tables

**Figure 1 nutrients-13-01404-f001:**
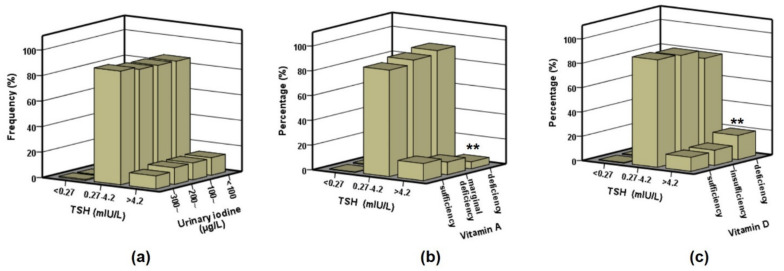
Distribution of TSH in different UIC, vitamin A and vitamin D groups. (**a**) TSH distribution in each UI group (<100, 100~, 200~, and 300~ µg/L); (**b**) TSH distribution in each VA group (deficiency, marginal deficiency, sufficiency); (**c**) TSH distribution in each VD group (deficiency, insufficiency, sufficiency). Values are percentage. Statistical differences to the sufficiency groups are shown as ** *p* < 0.01.

**Table 1 nutrients-13-01404-t001:** Characteristics of Chinese school-age children from three different regions (median, P_25_–P_75_).

Characteristics	Hebei (n = 1349)	Zhejiang (n = 1311)	Guangxi (n = 1148)	Total (n = 3808)
Boys(n, %)	682(50.56)	666(50.80)	581(50.61)	1929(50.66)
Girls(n, %)	667(49.46)	645(49.20)	567(49.39)	1879(49.34)
Age(years)	11.00(9.00–13.00)	11.42(8.87–13.76)	11.75(9.20–14.26)	11.32(9.00–13.97)
BMI(kg/m^2^)	18.87(16.47–21.61)	17.66(15.86–20.40)	16.856(15.20–19.07)	17.80(15.83–20.41)
UIC(μg/L)	180.90(123.00–234.30)	171.65(122.99–240.38)	210.10(149.03–296.50)	185.14(129.60–252.88)
UCr(μmol/L)	7842.00(4435.00–12762.50)	12,973.50(8515.09–18,894.50)	14,070.00(9028.75–20,076.50)	11,400.50(6890.50–17,279.84)
UI/Cr(μg/g)	197.24(144.64–271.44)	115.09(83.76–166.24)	135.45(100.33–188.10)	148.61(103.49–216.91)
TSH (mIU/L)	2.63(1.91–3.72)	1.87(1.30–2.60)	2.62(1.88–3.57)	2.36(1.64–3.28)
Vitamin A (μg/mL)	0.37(0.31–0.43)	0.33(0.26–0.40)	0.40(0.34–0.47)	0.37(0.30–0.44)
25 (OH) D (ng/mL)	15.40(11.00–20.90)	25.008(19.64–30.70)	22.40(19.10–26.20)	21.30(15.70–26.64)

**Table 2 nutrients-13-01404-t002:** Median and frequency distribution of UI among Chinese school-age children.

Factors	N	Median UIC, P_25_–P_75_ (μg/L)	Frequency Distribution (%) Per UI Range, μg/L				
			<100	100~	200~	300~	*p*
Region							<0.01
Hebei	1349	180.90(123.00–234.30)	218(16.16)	590(43.74)	451(33.43)	90(6.67)	
Zhejiang	1311	171.65(122.99–240.38)	202(15.41)	612(46.68)	319(24.33)	178(13.58)	
Guangxi	1148	210.10(149.03–296.50) ^a, b^	107(9.32)	428(37.28)	334(29.09)	279(24.30)	
Gender							<0.01
Boys	1929	183.60(130.95–245.45)	240(12.44)	884(45.83)	557(28.88)	248(12.86)	
Girls	1879	187.60(128.10–259.98)	287(15.27)	746(39.70)	547(29.11)	299(15.91)	
Age (y)							<0.01
6~	525	149.78(105.29–207.60)	118(22.48)	263(50.10)	109(20.76)	35(6.67)	
8~	788	164.60(115.50–219.83)	138(17.51)	393(49.87)	193(24.49)	64(8.12)	
10~	755	179.08(127.40–242.40) ^c, d^	104(13.77)	342(45.30)	214(28.34)	95(12.58)	
12~	794	210.15(150.19–278.45) ^e, f, g^	68(8.56)	305(38.41)	270(34.01)	151(19.02)	
14~	946	211.45(150.98–286.72) ^h, i, j^	99(10.47)	327(34.57)	318(33.62)	202(21.35)	
Body weight							<0.01
Underweight	290	180.45(131.40–251.13)	33(11.38)	138(47.59)	77(26.55)	42(14.48)	
Normal weight	2670	188.60(131.40–257.88)	364(13.63)	1101(41.24)	782(29.28)	423(15.84)	
Overweight/obesity	848	174.08(126.57–237.78) ^k^	130(15.33)	391(46.11)	245(28.89)	82(9.67)	
VA							<0.01
deficiency	126	164.45(126.80–225.89)	17(13.49)	68(53.97)	26(20.63)	15(11.90)	
marginal deficiency	769	176.69(124.00–239.05)	115(14.95)	350(45.51)	225(29.26)	79(10.27)	
sufficiency	2913	187.81(131.60–257.95) ^l^	395(13.56)	1212(41.61)	853(29.28)	453(15.55)	
VD							<0.01
deficiency	464	189.95(133.33–242.98)	53(11.42)	205(44.18)	163(35.13)	43(9.27)	
insufficiency	1197	191.70(136.29–255.45)	160(13.366)	482(40.267)	379(31.66)	176(14.70)	
sufficiency	2147	180.10(126.60–256.20)	314(14.63)	943(43.92)	562(26.18)	328(15.28)	
Total	3808	185.14(129.60–252.88)	527(13.84)	1630(42.80)	1104(28.99)	547(14.36)	

^a^: Hebei vs. Guangxi; ^b^: Zhejiang vs. Guangxi; ^c, d, e, f, g, h, i, j^: 6~ vs. 10~, 8~ vs. 10~, 6~ vs. 12~, 8~ vs. 12, 10~ vs. 12~, 6~ vs. 12~, 8~ vs. 12, 10~ vs. 12~; ^k^: normal vs. overweight; ^l^: deficiency vs. sufficiency. Statistical significance was considered when *p* < 0.05.

**Table 3 nutrients-13-01404-t003:** Median and distribution of TSH of Chinese school-age children (mIU/L).

Factors	N	Median (P_25_–P_75_)	Frequency Distribution (%) Per TSH Range			
			<0.27	0.27–4.20	>4.20	*p*
Region						<0.01
Hebei	1349	2.63(1.91–3.72)	4(0.30)	1091(80.87)	254(18.83)	
Zhejiang	1311	1.87(1.30–2.60) ^a^	10(0.76)	1241(94.66)	60(4.58)	
Guangxi	1148	2.62(1.88–3.57) ^b^	2(0.17)	970(84.49)	176(15.33)	
Gender						0.21
Boys	1929	2.40(1.65–3.34)	6(0.31)	1660(86.05)	263(13.63)	
Girls	1879	2.32(1.63–3.23)	10(0.53)	1642(87.39)	227(12.08)	
Age(years)						<0.01
6~	525	2.47(1.75–3.59)	5(0.95)	433(82.48)	87(16.57)	
8~	788	2.57(1.88–3.55)	0(0.00)	673(85.41)	115(14.59)	
10~	755	2.43(1.71–3.45)	2(0.26)	628(83.18)	125(16.56)	
12~	794	2.32(1.65–3.09) ^d^	3(0.38)	705(88.79)	86(10.83)	
14~	946	2.07(1.39–2.88) ^c, e, f, g^	6(0.63)	863(91.23)	77(8.14)	
Body weight						<0.01
Underweight	290	2.26(1.69–3.34)	0(0.00)	257(88.62)	33(11.38)	
Normal weight	2670	2.33(1.60–3.20)	15(0.56)	2335(87.45)	320(11.99)	
Overweight/obesity	848	2.51(1.73–3.57) ^h^	1(0.12)	713(84.08)	487(15.80)	
Total	3808	2.36(1.64–3.28)	16(0.42)	3305(86.79)	487(12.79)	

^a^: Hebei vs. Zhejiang; ^b^: Zhejiang vs. Guangxi; ^c, d, e, f, g^: 6~ vs. 14~, 8~ vs. 12~, 8~ vs. 14~, 10~ vs. 14~, 12~ vs. 14~; ^h^: normal vs. overweight. Statistical significance was considered when *p* < 0.05.

**Table 4 nutrients-13-01404-t004:** Characteristics of schoolchildren with underweight, normal weight, and overweight/obesity (median, P_25_–P_75_).

	Boys			Girls		
	Underweight (n = 173)	Normal Weight (n = 1254)	Overweight/Obesity (n = 502)	Underweight (n = 117)	Normal Weight (n = 1416)	Overweight/Obesity (n = 346)
UIC (μg/L)	180.20(140.69–246.15)	185.59(128.30–250.65)	176.00(131.05–232.08)	180.70(108.20–255.45)	192.87(134.48–265.43)	171.65(117.16–246.75) ^**^
TSH (mIU/L)	2.13(1.68–3.14)	2.37(1.60–3.22)	2.62(1.72–3.81)^ ##,**^	2.48(1.68–3.57)	2.30(1.60–3.17)	2.37(1.76–3.30)^c^
Vitamin A (μg/mL)	0.35(0.28–0.42)	0.36(0.30–0.43)	0.39(0.31–0.47) ^##, **^	0.34(0.29–0.41)	0.37(0.31–0.43)	0.37(0.30–0.44) ^c^
25 (OH) D (ng/mL)	23.49(19.10–28.90)	22.10(16.88–27.62)	22.27(15.43–27.72)	21.00(16.82–24.65) ^a^	20.30(15.10–25.60) ^b^	19.08(13.68–24.72) ^c^

^##^: overweight/obesity vs. underweight; ^**^: overweight/obesity vs. normal weight; ^a^: underweight boys vs. underweight girls; ^b^: normal weight boys vs. normal weight girls; ^c^: overweight/obese boys vs. overweight/obese girls. Statistical significance was considered when *p* < 0.05, ** or ^##^: *p* < 0.01.

**Table 5 nutrients-13-01404-t005:** Generalized linear model of the association between TSH and UIC, vitamin A/D status.

Factors	β (95% CI)	*p*
UIC (μg/L)		
<100	−0.066 (−0.253 to 0.122)	0.491
100–199	−0.010 (−0.162 to 0.142)	0.899
200–299	0.066 (−0.092 to 0.224)	0.413
≥300	0	
VA		
deficiency	−0.597 (−0.868 to −0.326)	0.000
marginal deficiency	−0.318 (−0.441 to −0.194)	0.000
sufficiency	0	
VD		
deficiency	0.609 (0.437 to 0.781)	0.000
insufficiency	0.225 (0.111 to 0.338)	0.000
sufficiency	0	

Model was adjusted for region, age, sex and BMI. Statistical significance was considered when *p* < 0.05.

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
