# Peer review of "Iodine Nutritional Status and Related Factors among Chinese School-Age Children in Three Different Areas: A Cross-Sectional Study"

_nutrients, 2021, doi:10.3390/nu13051404_

Round 1

Reviewer 1 Report

In this cross sectional study, the authors aimed to assess the iodine nutritional status of 3808 children (aged 6- 17 years) from three representative provinces of China evaluating urinary iodine concentration (UIC) and TSH. In addition, to investigate the relationship between UIC and TSH, body mass index (BMI), vitamin A and VitaminD were also measured. Although monitoring iodine status is an important assessment, I have some recommendations for revisions to improve the manuscript.

INTRODUCTION

The rationale for the study regarding the investigation of Vitamin A and Vitamin D should be strengthened.

MATERIALS AND METHODS

-Please describe anthropometric measurements and reference values for underweight, normal and overweight in children aged 6-17 years.

- In the analysis of data, effort should be made to control or adjust for such confounding variables, such as BMI, age and sex.

RESULTS

In table 1 values from girls should be included.

Table 4 should report data from linear model analysis controlled with potential confounders.

Figure 1 should be omitted.

DISCUSSION

I would recommend authors to discuss results from data analysed with confounders as well as to more adequately justify data of Vitamin A and D.

Minor Revisions

Please review minor grammar throughout the manuscript.

Reviewer 2 Report

Dear Authors,

    I have read with huge interest your manuscript entitled “Iodine Nutritional Status and Related Factors among Chinese School-age Children in Three Different Areas: A Cross-sectional Study” submitted to Nutrients.

This work offers a very comprehensive insight into the nutritional status of iodine, vitamin A and Vitamin D in a representative sample of school-age children from 3 different areas in China. The paper contains adequate tables and figures to illustrate the information provided. The discussion is clear and well written.

I only have a single comment: I miss additional information about the results of UIC, TSH, VA and VD in boys and girls according to BMI. Since a 22.06% of children were classified as overweighted, it would be desirable to see the significance of BMI and sex on iodine and vitamin levels. I suggest to add a figure or a new table to extend this information.

Round 2

Reviewer 1 Report

The authors have satisfactorily responded to all my questions and made the necessary changes to the manuscript.